# Edge computing task scheduling mechanism based on multi-dimensional feature extraction and attention fusion

**Shunli Zhang**[1,2]*, **Jia-ying Li**[1], **Peng Yu**[2]

**1** Department of Information Technology and Engineering, Jinzhong University, Jinzhong, People's Republic of China, **2** State key Laboratory of Networking and Switching Technology, Beijing University of Posts and Telecommunications, Beijing, People's Republic of China,

\* 544891117@qq.com

## Abstract

In the edge computing environment, when existing task scheduling algorithms allocate resources for tasks, the central host of edge computing consumes more energy and processes fewer tasks successfully. To solve this problem, this paper proposes an edge computing task scheduling mechanism based on multi-dimensional feature extraction and attention fusion (MFEAF). MFEAF achieves efficient fault prediction and fault-tolerant scheduling optimization by integrating graph attention network and temporal network modeling. In order to capture the dynamic dependency relationships between hosts, this paper adopts a multi-level graph neural network architecture that integrates graph convolution and graph attention mechanisms to extract features from the scheduling decisions and time state data of hosts. By dynamically adjusting the learning rate and cosine annealing strategy, redundant transfer is reduced and convergence efficiency is improved. The experimental results show that in terms of fault prediction performance, the F1 score of MFEAF reaches 0.9328. Compared with the latest method, the F1 score of our method has increased by 5.83%, the accuracy has improved by 9.27%, and the recall rate has increased by 2.46%. In terms of energy efficiency and task processing capability, the average energy consumption decreased by 5.0%, and the number of completed tasks increased by 12.0%. In terms of migration efficiency, the average migration time has been reduced by 50%, with a total migration time of only 19.79 seconds, a decrease of 51.3% compared to the suboptimal model. In terms of cost and fairness, containers have the lowest cost and the highest fairness index, reflecting the balance of resource allocation and high cost-effectiveness. In conclusion, MFEAF provides an efficient and adaptive solution for dynamic fault tolerance in edge computing environment.

**Data availability statement:** All relevant data are within the paper and its Supporting Information files.

**Funding:** This work was supported by Open Foundation of State key Laboratory of Networking and Switching Technology (Beijing University of Posts and Telecommunications) (SKLNST-2024-2-01), Jinzhong University Doctoral Research Funds (20210104),Shanxi Province Teaching Research and Reform Project (J20241337),2024 Shanxi Provincial Key Research and Development Program Projects (202402120101009).

**Competing interests:** The authors have declared that no competing interests exist.

# 1 Introduction

With the explosive growth of IoT devices, front-end sensors generate a large amount of data. It is not feasible to send all data to the cloud backend for processing, and edge computing technology came into being to meet the needs of local data processing [1]. However, the device resources in the edge computing environment are limited, which brings challenges to the reliability of services. When tasks arrive intensively or suddenly, it often leads to resource contention and system overload, causing application performance degradation and failures. Therefore, task scheduling in edge computing environment and edge computing resource reliability assurance mechanism have become current research focus. In the field of edge computing reliability assurance, existing research has focused on solving resource constraints and fault tolerance problems, specifically covering traditional fault tolerance methods, unsupervised learning applications, deep reinforcement learning exploration, and integration with other technologies.

In the aspect of traditional fault tolerance, the problem of edge computing fault tolerance is solved through heuristic or traditional supervised learning strategies. Sivagami V M et al [2] proposes the dynamic fault-tolerant virtual machine migration algorithm DFTM, which utilizes real-time virtual machine migration to address faults in cloud computing environments. However, its integer linear programming model has poor scalability in large-scale edge deployment due to state explosion. Liu J et al [3] proposes an active coordination fault-tolerant method PCFT, which uses particle swarm optimization to solve the coordination consensus problem in multi heterogeneous cloud node fault tolerance, to optimize network consumption, transmission overhead, and service response time. Liu G et al [4] proposes a load balancing algorithm, but faces challenges in selecting migration tasks and target host nodes. These traditional methods are difficult to deal with fault tolerance requirements efficiently and accurately in the complex and changeable edge computing environment.

Because unsupervised learning can process unlabeled data, it has great application potential in the field of edge computing fault tolerance. Previous studies have utilized unsupervised learning techniques such as sparse neural networks [5] and autoencoders [6] to extract features from historical log data for fault prediction. However, these methods are only suitable for identifying the latest network status and are not conducive to solving the problem of active fault recovery. Sharif A et al [7] proposes an efficient checkpoint and load balancing method ECLB for selecting tasks from overloaded hosts and target hosts from underloaded hosts. But this method does not consider other resource types such as memory, disk, and network.

In the exploration of deep reinforcement learning and intelligent algorithms, Li Y et al [8] combines the characteristics of deep reinforcement learning (DRL) and edge computing to achieve more efficient fault tolerance strategies. However, this method faces problems such as complex training, slow convergence speed, and difficulty in adapting to dynamic environmental changes. Tuli S et al [9] proposed the DeepFT model, which can predict and diagnose faults in the system. However, the DeepFT model has not fully explored the features of network resources and the correlation

between features. With the development of deep learning technology, various neural networks are being jointly designed and applied by more researchers, achieving good results in solving practical problems. Rao N V et al [10] designed an attention based RNN architecture to improve the performance of modulation classification algorithms in cognitive radio networks. Shen L et al [11] designed a CNN-GRU deep learning framework to improve the performance of debris flow velocity prediction. Chiang W L et al [12] designed Graph Convolutional Network (GCN) with greater depth and breadth. Wang X et al [13] designed a management network for graphic attention features based on visual features. Aghapour Z et al [14] divides the CNN network into multiple sub networks based on the requirements of task offloading. Deng X et al [15] applied deep reinforcement learning (DRL) to adaptive resource allocation algorithms in dynamic environments, improving the fairness of task allocation and the load balancing characteristics of resources. Rathor V S et al [16] designs a sparse CNN and a dedicated indexing module, significantly improving hardware energy efficiency with a slight loss in accuracy. Behera S R et al [17] utilizes time series prediction for edge resources and optimizes task allocation, enhancing the efficiency and energy efficiency of mobile edge computing. The results obtained from these research methods provide some clues and more confidence for the network model design in this article.

Through analysis of existing research, it is known that good results have been achieved. However, when existing task scheduling algorithms are used to schedule tasks, edge computing center hosts consume more energy and have fewer tasks successfully processed. To solve this problem, this paper proposes an edge computing task scheduling mechanism based on multi-dimensional feature extraction and attention fusion (MFEAF). The core contributions of this paper can be summarized as follows:

(1) A multi-dimensional feature extraction architecture that integrates graph attention networks with temporal networks is proposed. For the first time, the combination of graph attention networks and graph convolutional networks is achieved, and LSTM and self-attention mechanisms are introduced to jointly model the dynamic dependencies between edge nodes, thereby significantly improving the accuracy of system state and fault prediction.

(2) Designed a training optimization mechanism incorporating adaptive learning rate and cosine annealing strategy. By dynamically adjusting the learning rate and incorporating the cosine annealing strategy, we effectively reduce the transmission of redundant information during training, accelerate model convergence, and enhance the adaptability and stability of the algorithm in dynamic edge environments.

(3) Realized end-to-end joint optimization of fault prediction and fault-tolerant scheduling. MFEAF not only achieves efficient fault prediction, but also outperforms existing benchmark methods in multiple dimensions such as energy efficiency, task completion, and migration cost through an integrated scheduling optimization algorithm.

Based on these innovations, MFEAF can effectively address issues such as low task execution success rates and high host energy consumption caused by limited device resources in edge computing environments, thereby enhancing the stability and resource utilization of edge computing systems.

## 2 Problem description

Suppose there are p IoT terminals, which need to send some services to the proxy node of the edge computing center when processing services. After receiving the task processing request, the agent node in the edge computing center schedules the task based on the utilization of the CPU, memory, hard disk, and network of each host in the edge computing center, with the optimization goal of maximizing the number of task processing and minimizing the energy consumption of the edge computing center host. The edge network includes a set of host nodes $H = \{h_1, ..., h_N\}$, with different dimensions such as CPU cores, memory capacity, and disk space for each host node. The system running time is a finite interval T, and the timeline is divided into equally long scheduling intervals $\{I_t\}_{t=0}^{T}$. Within each interval, the proxy node updates the scheduling strategy based on the latest host status.

Assuming that host node failures are mainly caused by resource contention, the probability of host node failures is positively correlated with resource utilization. Define a dynamic threshold function $\tau_h^{(c)}(t)$ to represent the utilization threshold of resource $c \in \{CPU, RAM, Disk\}$ by host h at time t. When the resource utilization rate $u_h^{(c)}(t) > \tau_h^{(c)}(t)$ is reached, it is marked as a resource contention failure. Threshold $\tau_h^{(c)}(t)$ is dynamically adjusted based on historical load and node performance. For example, it can be set based on the average utilization rate of k windows in the past. This article assumes that the node power supply is stable, and fault recovery only requires resource release or task migration, so non recoverable faults such as power outages can be ignored.

In terms of workload model, the task bag workload model is used to simulate edge computing scenarios. At the beginning of each scheduling interval, IoT devices generate a batch of parallelizable independent tasks and send them in bulk to edge agents. Task attributes include computing resource requirements, SLO deadlines, etc. Abstracting tasks into container instances to achieve resource isolation and rapid deployment. SLO constraint is the time limit for tasks from reaching the edge agent to completing processing.

At the beginning of scheduling interval $I_t$, the agent generates task scheduling decision $S_t \in \{0, 1\}^{p \times m}$, where p is the number of active tasks and $S_t(i,j) = 1$ represents that task i is assigned to host $h_j$. For new tasks, $S_t$ is the initial scheduling. For existing tasks, if $S_t(i,j) \neq S_{t-1}(i,j)$, it triggers the migration from the original host to the new host. In terms of state prediction, based on the historical state sequence $\{x_0, ..., x_t\}$, a sliding window $W_t = \{x_{t-k+1}, ..., x_t\}$ is used to predict the next state $x_{t+1}$. $x_t$ includes resource utilization indicators for each host.

## 3 Models

### 3.1 Model architecture

This paper designs a edge computing Task Scheduling Mechanism based on Multi-dimensional Feature Extraction and Attention Fusion (MFEAF). The model architecture of this mechanism is shown in Fig 1. This architecture achieves accurate prediction of system state and efficient fault-tolerant scheduling by deeply integrating temporal and graph structured data. The model architecture diagram includes the graph data preprocessing module, graph network processing module, LSTM and self-attention module, residual module, encoder module, state decoder module, and prototype decoder module.

The graph data preprocessing module includes obtaining the number of nodes, creating node pairs, removing self-loops, and generating graph networks. This module converts the original temporal multidimensional feature data into a graph structure representation, preparing for subsequent GCN and GAT processing. By creating node pairs to construct adjacency relationships and removing self-loops, the rationality of the graph network is ensured. The graph network processing module includes GCN and GAT. GCN network extracts node features through adjacency matrix and learns local structural information of the network. The GAT network introduces attention mechanism to allocate differentiated weights to neighboring nodes, enhancing the perception ability of important neighbors. This module can extract structured information between nodes and enhance node feature representation capabilities using adjacency information.

LSTM and self-attention modules include LSTM layers and self-attention mechanisms. The LSTM layer handles temporal dependencies. The self-attention mechanism implements dynamic weighting of key time steps to enhance the ability to capture long-term dependencies. This module implements the function of jointly modeling graph structures and temporal features. The residual module includes residual connections and residual mappings. Residual connections establish a direct path from input to output. Residual mapping learns the difference between input and output. This module alleviates gradient vanishing during deep model training, allowing information to propagate deeply and improving learning efficiency.

The encoder module includes flattening operation, fully connected layer LeakyReLU. The flattening operation converts the output of the graph structure into a vector. Fully connected layers and LeakyReLU perform nonlinear transformation and dimensionality reduction on graph convolution outputs. This module compresses graph embeddings into low dimensional latent vectors while preserving key information. The state decoder module includes a linear layer, LeakyReLU

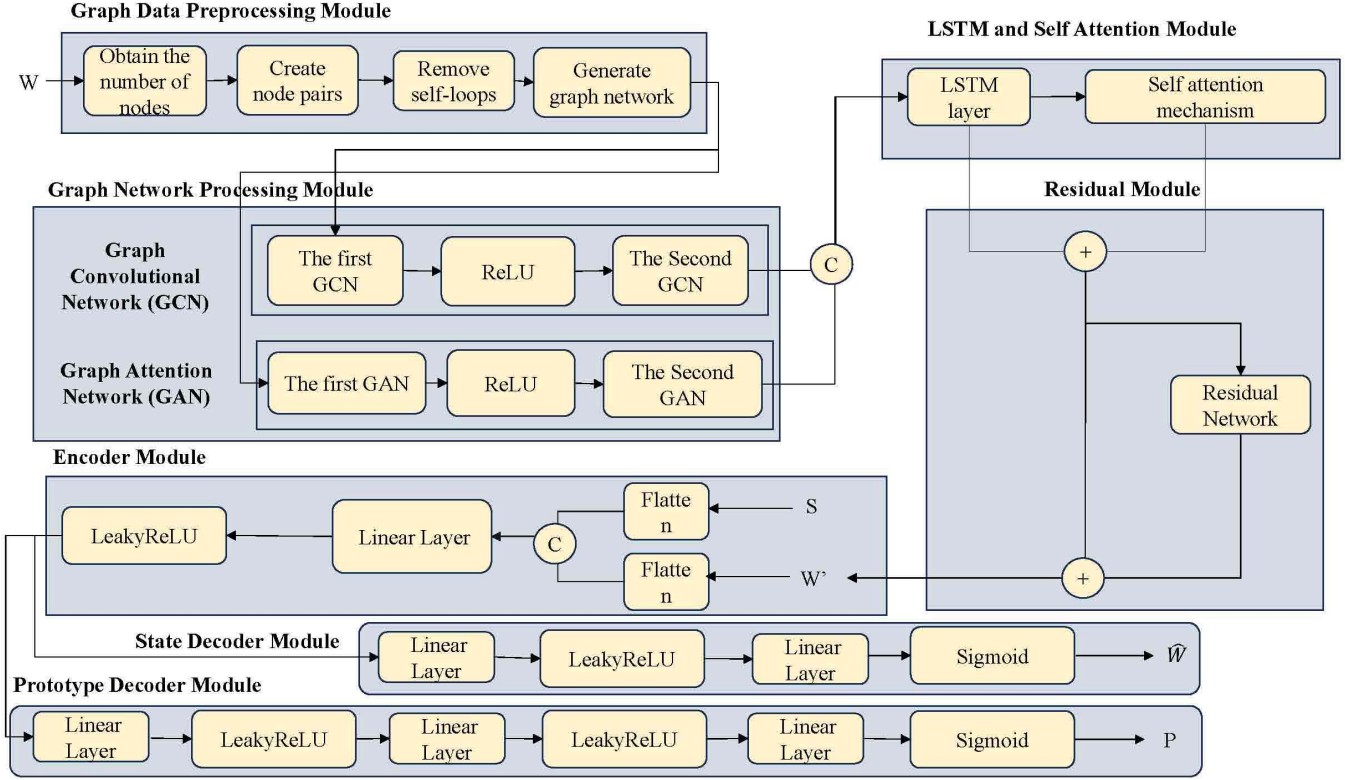

**Fig 1. Model architecture.**

activation function, linear layer, and Sigmoid activation function. This module restores the predicted state output from the embedded state, which is used to predict the state of the node in the next time window. The prototype decoder module includes a linear layer, LeakyReLU activation function, linear layer, LeakyReLU activation function, linear layer, and Sigmoid activation function. This module is used to calculate the distance between the input and each prototype to determine its prototype category.

### 3.2 Data preprocessing module

Convert raw time-series data and scheduling data into a format that the model can process, thereby enhancing feature representation capabilities. In terms of temporal data, the shape of the CPU, memory, and disk utilization sequence for each host is $[T, 3 \times N]$, T is the time step, and N is the number of hosts. In terms of scheduling data, the shape of the allocation relationship matrix between containers and hosts is $[T, N \times N]$.

In terms of time series data standardization, the time series is divided into fixed length windows, and when there are insufficient windows, the first frame data is used to fill in, ensuring consistent input dimensions. Normalize each feature dimension to $[0, 1]$ using formula (1). Here, $X_{train}$ represents the training dataset, $max(X_{train}, axis = 0)$ represents the maximum value of each feature dimension in the calculated training data, and $\epsilon$ represents the minimum value (such as 1e-8) to prevent division by zero.

$$X_{norm} = \frac{x}{max(X_{train}, axis = 0) + \epsilon}$$

(1)

In terms of scheduling data encoding, the scheduling decision matrix is transformed into a tensor as the static feature of the model input. The scheduling data represents the task allocation relationship between hosts in the form of an adjacency matrix, denoted as $S \in R^{N \times N}$, where N is the number of hosts.

In terms of anomaly threshold calculation, thresholds for each feature dimension are dynamically generated based on the percentile of the training data and calculated using formula (2). Among them, *data* represents the training dataset, *PERCENTILES* is the percentile (e.g., 98), indicating that the higher value of the data distribution is taken as the benchmark. *percentile_multiplier* is an adjustment coefficient (such as 0.99999) used to amplify the threshold to reduce false positives.

$$thresholds = np.percentile\,(data, PERCENTILES) \times percentile\_multiplier \qquad (2)$$

The adjustment coefficient is used to slightly lower the threshold to reduce false positives. Selection instructions for this value: (1) Numerical stability: When the data approaches the standardized upper limit, 0.99999 can slightly lower the threshold to avoid abnormal behavior caused by numerical accuracy issues. (2) The principle of minimum intervention: 0.99999 is very close to 1, and only minor adjustments are made to data that is close to saturation, without significantly changing the overall threshold distribution. (3) Balanced detection sensitivity: Combined with the subsequent 0.7 times threshold adjustment, a two-level threshold adjustment mechanism is formed, which ensures both numerical stability and sufficient anomaly detection sensitivity. Through this design, a good balance is achieved between ensuring numerical stability and detection performance.

### 3.3 Module design

**3.3.1 Graph data preprocessing module.** This module aims to provide structural prior information for graph neural networks, namely the connection topology between hosts, and represent it as an adjacency matrix $A \in R^{N \times N}$. A represents the connection relationship between nodes without self-loops. D is a degree matrix that records the number of neighbors for each node. $A_{ij} = 1$ indicates that there is an edge connection between node i and node j, otherwise it is 0. This structure avoids redundant self-connections, which is beneficial for enhancing the modeling ability of inter node interactions and improving the robustness of spatial dependency modeling.

$$A_{ij} = \begin{cases} 1, i \neq j \\ 0, i = j \end{cases} \qquad (3)$$

The removal of self-loops in this study is an application specific architectural decision aimed at better balancing the capture of relationships between nodes and the representation of node characteristics. Although standard GCNs typically add self-loops to preserve node self-information, in complex architectures such as MFEAF with multi-component fusion, residual connections and multi-layer networks can effectively capture and preserve node self-features. Meanwhile, it was found in the experiment that removing self-loops can improve the modeling ability of the model for inter node dependencies in edge task scheduling scenarios.

**3.3.2 Network processing module.**

1)   Graph convolution layer

Based on GCN to aggregate neighbor node features, the GCN consists of two stacked layers, both updated according to the following formula (4). The output of each layer serves as the input for the next layer. $W_{GCN}$ represents the trainable weight matrix of this layer. $H_{GCN}^{(l)}$ is the input feature of layer $l$, where $l = 0$ represents the first layer and the input is $W_t$. $l = 1$ represents the second layer and the input is the output $H^{(1)}$ of the previous layer. $D^{-\frac{1}{2}} A D^{-\frac{1}{2}}$ normalizes the adjacency matrix A to prevent large differences in node degrees from causing numerical instability during aggregation,

ensuring that neighbor information evenly affects node features. After two layers of graph convolution, structural enhancement feature $H_{GCN} \in R^{N \times d_{latent}}$ is obtained for fusion with the GAT module results. $d_{latent}$ represents the dimension of the hidden layer.

$$H_{GCN}^{(l+1)} = ReLU\left(D^{-\frac{1}{2}} A D^{-\frac{1}{2}} H_{GCN}^{(l)} W_{GCN}\right)$$

(4)

2) Graph Attention Layer

The GAT consists of two stacked layers, each layer using the following attention computing mechanism. Among them, $i$ and $j$ represent two nodes in the graph. $N_i$ represents the set of neighboring nodes of node $i$. $h_i$ represents node features with a dimension of F. $W_{gat}$ represents the weight matrix, which linearly transforms the features of each node into a new space. $\|$ represents the feature concatenation operation. $W_{gat}h_i$ represents the feature of node $i$ after linear mapping transformation. $a$ represents the attention vector, which maps the concatenated features into a scalar attention score. $a^T[W_{gat}h_i \| W_{gat}h_j]$ represents the attention score, indicating the correlation between $i$ and $j$. $exp(.)$ represents an exponential function, using softmax operation to achieve positive and normalized attention weights. $\alpha_{ij}$ represents the normalized attention weight, indicating the degree of attention that node $i$ pays when aggregating features from neighbor $j$.

$$\alpha_{ij} = \frac{\exp\left(LeakyReLU\left(a^T[W_{gat}h_i \| W_{gat}h_j]\right)\right)}{\sum_{k \in N_i} \exp\left(LeakyReLU\left(a^T[W_{gat}h_i \| W_{gat}h_k]\right)\right)}$$

(5)

The weighted aggregation of node features is shown in formula 6. Where $l$ represents the number of GAT layers. $\sigma(.)$ represents the nonlinear activation function. GAT uses attention mechanism to weight and sum the features of neighboring nodes. The representation of node $i$ in layer $l+1$ is obtained by linearly transforming its neighbor features $h_j^{(l)}$ and weighting them with attention weights $\alpha_{ij}^{(l)}$, and then summing them up. After calculating the activation function, the importance of different neighbors is dynamically modeled. By using two layers of attention calculation, output structure enhanced feature $H_{GAT} = \left\{h_i^{(2)}\right\} \in R^{N \times d_{latent}}$ for fusion with GCN layer results.

$$h_i^{(l+1)} = \sigma\left(\sum_{j \in N_i} \alpha_{ij}^{(l)} W_{gat}^{(l)} h_j^{(l)}\right)$$

(6)

**3.3.3 LSTM and self-attention module.** To model the trend of host resource evolution over time, this module uses Long Short Term Memory (LSTM) networks to encode the structural embedding sequences extracted by GCN and GAT. The model first fuses the host features processed by the graph neural network at each time step. Then stack the fused features of multiple time steps into a sequence input.

$$H^{(t)} = H_{GCN}^{(t)} + H_{GAT}^{(t)}$$

(7)

$$H = \left[H^{(1)}, H^{(2)}, ..., H^{(T)}\right]$$

(8)

LSTM can effectively capture long-term dependencies in sequences, representing the graph structure $H_i^{(1)}, ..., H_i^{(T)}$ of each host node $i$ at time steps, and stacking the temporal features of all nodes into sequence $t_{combined} \in R^{T \times d_{latent}}$ according to time steps. To adapt to the PyTorch LSTM input format, the batch dimension is extended to make the input $R^{1 \times T \times d_{latent}}$ (formula 9), which is then fed into the LSTM module. The output $H_{lstm} \in R^{1 \times T \times d_{latent}}$ is obtained by removing the batch

dimension to obtain a temporal encoding representation (Formula 10). This article takes the output of the last time step as the temporal feature representation of host $i$.

$$H_{lstm} = LSTM\left(t_{combined}.unsqueeze\left(0\right)\right) \tag{9}$$

$$H_{lstm} = H_{lstm}.squeeze\left(0\right) \in R^{T \times d_{latent}} \tag{10}$$

In order to further model the global dependency relationship between time steps, the model introduces a self-attention mechanism. This mechanism dynamically adjusts the weight distribution of key dimensions by calculating the feature correlation between each time step. $H_{attn} \in R^{T \times d_{latent}}$ represents the temporal feature representation after attention enhancement, used to capture important information dependencies across time steps. This module can dynamically identify and reinforce key time periods, improving the model's ability to perceive anomalies or unexpected events.

$$H_{attn} = MultiHeadAttention\left(H_{lstm}, H_{lstm}, H_{lstm}\right) \tag{11}$$

**3.3.4 Residual module.** To enhance the modeling ability of the model for key events in time series, this module introduces a dual residual connection structure based on temporal feature encoding. Firstly, after obtaining the output sequence $H_{lstm} \in R^{T \times d_{hidden}}$ of LSTM, the model extracts the global dependency relationship between time steps through Multi Head Attention and outputs the attention enhanced feature $H_{attn} \in R^{T \times d_{hidden}}$. $d_{hidden}$ represents the dimension of the hidden layer. Then, the two are connected by residual connection. $T_{res-attn} \in R^{T \times d_{hidden}}$ is the enhanced sequence feature after residual connection.

$$T_{res-attn} = H_{lstm} + H_{attn} \tag{12}$$

**3.3.5 Encoder module.** After extracting spatiotemporal features, the model uses an encoder decoder structure to reconstruct the system state and compresses it into a low dimensional representation of the latent space for anomaly detection and reconstruction. Input $T_{res-attn}$ into the linear transformation layer and add it to itself to form the final temporal enhanced representation. $W_{res} \in R^{d_{hidden} \times d_{hidden}}$ is the residual fully connected layer weight.

$$T_{residual} = T_{res-attn} + W_{res} . T_{res-attn} \tag{13}$$

Flatten $T_{residual}$ into a one-dimensional vector and concatenate it with the flattened scheduling data $S \in R^{N \times N}$ as input to the encoder.

$$z_{in} = Flatten\left(T_{residual}\right) || Flatten\left(S\right) \tag{14}$$

The encoder consists of two layers of linear mapping and LeakyReLU activation function, which are used to perform non-linear compression on the concatenated high-dimensional feature vectors to obtain the latent representation $z$. $W_1$ and $W_2$ represent the fully connected layer weight. $b_1$ and $b_2$ is the corresponding bias term. $z$ represents the potential state of the final output.

$$z = LeakyReLU\left(W_2 . LeakyReLU\left(W_1 . z_{in} + b_1\right) + b_2\right) \tag{15}$$

**3.3.6 State decoder module.** This module is used to restore the current system state from the latent representation $z \in R^{d_{latent}}$, that is, to predict the normalized values of each indicator. The first layer of nonlinear transformation is activated using LeakyReLU. The second layer is mapped and activated through Sigmoid to output the predicted state.

$W_1 \in R^{d_{hidden} \times d_{latent}}$ and $b_1$ represent the first linear layer parameter. $W_2 \in R^{3 \times d_{hidden}}$ and $b_2$ represents the second linear layer parameter. $\hat{W} \in [0, 1]^3$ represents the normalized system state, with three dimensions being CPU, memory, and disk utilization.

$$h_{state} = LeakyReLU(W_1 \cdot z + b_1) \tag{16}$$

$$\hat{W} = \sigma(W_2 \cdot h_{state} + b_2) \tag{17}$$

**3.3.7 Prototype decoder.** To achieve the discrimination of abnormal categories, $K + 1$ prototype vectors are generated from the latent representation using a multi-layer perceptron. The prototype decoder consists of three layers of linear mapping, LeakyReLU and Sigmoid activation function, which are used to perform nonlinear compression on the concatenated high-dimensional feature vectors to obtain latent state representations $h_1$, $h_2$, and $P_k$, respectively. $W_{P1}$, $W_{P2}$ and $W_{P3}$ represent fully connected layer weights. $b_{p1}$, $b_{p2}$ and $b_{p3}$ are corresponding bias terms.

$$h_1 = LeakyReLU(W_{P1} \cdot z + b_{p1}) \tag{18}$$

$$h_2 = LeakyReLU(W_{P2} \cdot h_1 + b_{p2}) \tag{19}$$

$$P_k = \sigma(W_{P3} \cdot h_2 + b_{p3}) \tag{20}$$

# 4 Algorithms

To improve the performance of MFEAF, this paper designs a model training algorithm and a scheduling optimization algorithm.

## 4.1 Model training algorithm

The model training algorithm constructs a robust fault-tolerant recovery mechanism by jointly optimizing the model's state prediction ability and anomaly classification ability. The model training algorithm includes spatiotemporal encoding of historical state features and scheduling information, decoding of states and prototypes, calculation of loss functions and parameter optimization, dynamic prototype updates, and adaptive adjustment of update factors.

**Algorithm 1. The model training algorithm**
**Input:** Initial model, historical time window host state feature $W_t$, scheduling decision matrix $S_t$, category prototype set $P = \{p_k\}_{k=1}^{K}$, dynamic anomaly threshold $\tau$
**Output:** Trained model
1: Traverse the training sample sequence to obtain historical status input $W_t$ and current scheduling matrix $S_t$
2: Input $W_t$ and $S_t$ into the model;
3: The model outputs the predicted resource state $\hat{W}_t$ for the next moment and the set of embedding vectors $Z$ for the current host;
4: Calculate the mean square error between the predicted state and the true state, denoted as state reconstruction loss $L_{rec}$ in Eq. (22);
5: Initialize triplet loss $L_{triplet}$;
6: for Obtain prototypes for each host $p_k$:
7:     Determine whether its predicted state exceeds the threshold $\tau$ in Eq. (23);

```
8:     According to the judgment result, consider the host as a "normal class" or "abnormal class";
9:     Select the corresponding prototype to calculate the triplet loss in Eq. (25):
10:        If it is an abnormal class, select the nearest abnormal prototype as the positive class;
11:        Otherwise, it is considered a normal class and the normal prototype is selected as the
           positive class;
12:     If the current triplet satisfies the update condition (positive class loss ≤ minimum loss of
        all negative classes), then update the prototype of that class in Eq. (27);
13: Calculate the total loss in Eq. (26);
14: If there is a prototype update in this round of training, the prototype update factor will be
    attenuated in Eq. (28);
15: Return the trained model
```

**(1)  Data input (lines 1–2) and module prediction (line 3)**

In the training process, the historical time series feature window $W_t$ and the current scheduling decision matrix $S_t$ are used as inputs. Through the spatiotemporal feature extraction module, the next time resource state prediction value $\hat{W}_t$ and each host embedded representation set $Z$ are output. The embedded vector is distance matched with the prototype set $P = \{p_k\}_{k=0}^{K}$ to build a loss function. Among them, $p_0$ represents the normal class prototype, and $p_1 \sim p_K$ represents the abnormal class prototype.

$$Z = \{z_i\}_{i=1}^{N} \tag{21}$$

**(2)  Loss function**

**a)  State reconstruction loss (line 4)**

State reconstruction loss is the mean square error between the predicted state $\hat{x}_i$ and the true state $x_i$ of the computational model, used to measure prediction accuracy. Among them, $N$ is the number of hosts.

$$L_{rec} = \frac{1}{N}\sum_{i=1}^{N} \left|\left|\hat{x}_i - x_i\right|\right|^2 \tag{22}$$

**b)  Triple loss (line 5)**

Perform anomaly detection on predicted state $\hat{x}_i$ and resource threshold $\tau_i$ to determine the category of the sample (lines 6–8). This judgment result is used to construct the positive and negative class prototype selection strategy in triplet loss. For each anchor sample $z_k$, its positive prototype $p^+$ corresponds to the prototype vector of the anchor's category, and its negative prototype $p^-$ is the prototype vector of other categories.

$$\delta_i = \begin{cases} 1, if \ \hat{x}_i > \tau_i & \text{Abnormal} \\ 0, \ otherwise & \text{Normal} \end{cases} \tag{23}$$

The triplet loss is defined as follows, used to optimize the structure of the embedding space, pushing the anchor feature vector closer to the positive class prototype and farther away from the negative class prototype (lines 9–11). Among them, $\gamma$ is the margin, which controls the minimum interval. $p^+$ is a positive prototype, representing the prototype vector of the category to which the anchor sample belongs. Use the mean square error between anchor point and $p^+$ to represent the positive class loss. The smaller the positive class loss, the closer the anchor feature is to the positive class prototype. $p^-$ is the negative prototype, representing the prototype vectors of other categories. Use the mean square error between

anchor point and $p^-$ to represent negative class loss. The greater the negative class loss, the farther the anchor feature is from the negative class prototype.

$$L_{triplet}^{(k)} = max\left(0, ||z_k - p^+||^2 - ||z_k - p^-||^2 + \gamma\right)$$
(24)

c) Total loss (line 13)

Weighted average triplet loss $L_{triplet}$ enhances the ability to distinguish abnormal classes. Among them, $z_i$ is the embedded feature vector of the i-th host. Based on the above two losses, construct the total loss $L_{total}$ for the overall optimization objective. $\lambda$ is the balance hyperparameter used to balance the reconstruction accuracy and the proportion of prototype distribution optimization.

$$L_{triplet} = \frac{1}{N}\sum_{i=1}^{N} max\left(0, ||z_i - p^+||^2 - ||z_i - p^-||^2 + \gamma\right)$$
(25)

$$L_{total} = L_{rec} + \lambda \cdot L_{triplet}$$
(26)

The following discusses whether there is a cyclic dependency in triplet loss and how to avoid confirmation bias. The reason why this design does not have the problem of circular dependencies includes: abnormal labels are generated based on raw data and predefined thresholds before model training, not based on the predicted results of the model. The prototype vector will be updated based on sample features during the training process, but this is an iterative optimization process, not a cyclic dependency. Measures to avoid confirmation bias in this design include separating label generation from model training to ensure that labels are not affected by model predictions. By multiplying the threshold by 0.7, the detection sensitivity is increased to avoid confirmation bias caused by excessive reliance on strict thresholds. The prototype vector will be dynamically adjusted according to the training process, allowing the model to adapt to changes in data distribution rather than being fixed.

(3) Prototype update and parameter optimization (line 12)

When the positive class loss is minimized, the corresponding class prototype is updated using an attenuation factor. Among them, $\eta$ is the update weight, which is a dynamic decay update factor that determines the impact weight of new features on the prototype during the update. $z_{anchor}$ represents the feature vector of the current sample. $P_k^t$ is the vector of the $k$ -class prototype at time step $t$.

$$P_k^{t+1} = \eta \cdot z_{anchor} + (1-\eta) \cdot P_k^t$$
(27)

The calculation of $\eta$ is shown in formula (28) (line 14). Among them, $\eta^t$ represents the update factor used during the $t$ -th update, $\eta^{t-1}$ represents the update factor during the previous update, and *PROTO_FACTOR_DECAY* is a constant less than 1 used to control the attenuation of $\eta$.

$$\eta^t = \eta^{t-1} \cdot PROTO\_FACTOR\_DECAY$$
(28)

## 4.2 Scheduling optimization algorithm

The optimization process is executed independently in the reasoning phase, and does not update the model parameters. It is only used to modify and enhance the current scheduling strategy. The scheduling optimization algorithm takes the historical time window feature $W_t$ and the current scheduling matrix $S_t$ as inputs, and uses the model to perform the scheduling policy adjustment process under the parameters of resource threshold $\tau$, maximum iterations $T_{max}$, and continuous unchanged rounds threshold $T_{stop}$.

**Algorithm 2 The scheduling optimization algorithm**

**Input:** current historical time window $W_t$, current scheduling matrix $S_t$, model, host resource usage threshold $\tau$, maximum iterations $T_{max}$, termination count threshold $T_{stop}$

**Output:** Optimized scheduling strategy

1: The current scheduling matrix $S_t$ is converted into a trainable tensor $S_{init}$ and set as the current optimal scheme $S_{best}$;

2: Initialize AdamW optimizer and cosine annealing learning rate scheduler;

3: For maximum iteration times $T_{max}$:

4: Input scheduling variable $S_{init}$ and feature sequence $W_t$ into the model;

5: Extracting node spatial features through graph structure encoding;

6: Perform temporal modeling and enhance attention mechanism representation;

7: Encoder output fusion representation;

8: The decoder generates predicted resource state $\hat{W}_t$ and fault prototype P;

9: Calculate optimization loss $L_{opt}$ in Eq. (29);

10: If there is no abnormality in the predicted state (i.e., $L_{opt} = 0$), the optimization will be terminated immediately;

11: Clear the gradient and use backpropagation mechanism to update the learning rate;

12: Perform scheduling matrix projection;

13: Project the softmax output into a one hot format scheduling matrix;

14: Compare the current scheduling scheme with the last round of optimal scheme:

15: If the current scheduling scheme is the same as the previous round, increase the count of "unchanged rounds";

16: If the current scheduling scheme is different from the previous round, reset the "number of unchanged rounds" to zero;

17: If the number of unchanged rounds exceeds the set threshold $T_{stop}$, or if the current state is completely normal, terminate;

18: Return the current optimal scheduling solution as the final optimization result.

(1) Input parameters and initialization (lines 1–2)

Convert the scheduling matrix $S_t$ into a trainable tensor $S_{init}$ and set the initial optimal scheduling scheme $S_{best} = S_{init}$. Initialize the AdamW optimizer and cosine annealing learning rate scheduler, and set the unchanged round counter to 0.

(2) Model feature fusion (lines 4–8)

Combine $S_{init}$ and $W_t$ as model inputs. In each iteration, the model processes the input data according to the following steps. Firstly, graph structure encoding. Encode host connectivity relationships through GCN and GAT to capture spatial dependencies between nodes. Secondly, temporal encoding. Processing windowed temporal data through LSTM layers, capturing temporal dependencies, outputting hidden states, and enhancing feature expression through residual connections and self-attention mechanisms. Finally, encoding and decoding. The encoder concatenates spatiotemporal features with scheduling data and maps them to latent space, while the decoder generates predicted resource state $\hat{W}_t$ and fault prototype $P$.

(3) Optimization objective and loss function (lines 9–10)

Construct an optimization loss $L_{opt}$ with the goal of minimizing the number of anomalies exceeding the threshold in the predicted state, where $\tau_i$ A is the threshold for each dimension. If $L_{opt} = 0$, it indicates that the current scheduling has eliminated all exceptions, terminated the iteration in advance and returned to the optimal solution.

$$L_{opt} = \frac{1}{N} \sum_{i=1}^{N} ReLU\left(\hat{x}_i - \tau_i\right)$$

(29)

(4) Gradient optimization and scheduling projection (lines 11–17)

In the backpropagation and parameter update steps, AdamW optimizer is used to perform gradient descent, combined with cosine annealing learning rate scheduler to dynamically adjust the learning rate, and gradient pruning is used to ensure training stability (line 11). The initial learning rate is set to 0.8, and the cosine annealing learning rate scheduling period is set to A. After each iteration, the scheduling matrix S is projected back into one hot space to ensure that the final scheduling scheme is executable. Through the above optimization process, the final optimized scheduling plan is obtained.

$$S_{optimized} = arg \min_S L_{opt}(S)$$

(30)

Below is a detailed introduction to the AdamW optimizer, learning rate scheduler, and other components in the model. In the AdamW optimizer configuration, the model training optimizer lr = 0.0005 is used to update the parameters of the deep learning model, including the weights and biases of the neural network. Decision optimization optimizer lr = 0.8, used to optimize container scheduling decisions and find the optimal container allocation scheme, rather than training model parameters. In the configuration of the learning rate scheduler, the warm-up period is set to 10 epochs. The gradient clipping threshold is 1.0. The model learning rate is lr = 0.0008, which is the default learning rate attribute defined in the model class and is mainly used for default settings during model initialization and loading. The weight decay value is 1e-4. When traversing the entire dataset, batch size is trained on a sample-by-sample basis.

This high learning rate is reasonable in decision optimization scenarios for the following reasons: (1) These high learning rates are not used for model weight training, but for optimizing scheduling decision variables. (2) Container scheduling is a discrete problem that requires rapid search for feasible solutions in a continuous optimization space. (3) Use cosine annealing learning rate scheduler to ensure a rapid decrease in learning rate from 0.8. Therefore, the high learning rate of 0.8 used in this study is specifically designed for container scheduling decision optimization tasks, combining cosine annealing scheduler and discretization steps to make it both effective and stable in this specific scenario. This setting is different from traditional neural network training and is a special strategy adopted to solve optimization problems in discrete decision spaces.

In the One Hot encoding projection step, the optimized continuous value scheduling matrix is transformed into a probability distribution through softmax, and then projected into one hot encoding. Only retain the host allocation with the highest probability to ensure the discreteness and feasibility of the decision (lines 12–13). In the convergence judgment step, if there is no change in the continuous $T_{stop}$ -round scheduling matrix ($S_{new} = S_{best}$), it is considered as optimized convergence; Otherwise, update $S_{best}$ and reset the count (lines 14–17).

## 5 Performance analysis

### 5.1 Experimental setup

The experimental environment is Windows 10 operating system, 11th generation Intel Core i7-11800H processor, 16.0GB memory, and 4GB discrete graphics card. In terms of software, Python 3.8 is used as the development language, relying on the main libraries including graph neural network library DGL 2.2.1, scientific computing library NumPy 1.24.4, data visualization library Matplotlib 3.7.5, classic machine learning library scikit-learn1.3.2, deep learning framework PyTorch 2.2.1, and graph deep learning library torch geometry (PyG) 2.6.1 based on PyTorch. Time series data records the CPU, memory, and disk resource usage of the host, with a shape of (n_timesteps, 3 * n_hosts). The scheduling data represents the adjacency matrix of container allocation relationships, with the shape of (n_samples, n_hosts * n_hosts).

Fault labels are generated based on a dynamic threshold function (see formula 2). When the utilization of any resource dimension (CPU, memory, disk) exceeds this threshold, it is marked as a fault state. The threshold is dynamically adjusted based on the 98% percentile of historical data, with an adjustment coefficient of 0.99999, to reduce false positives.

## 5.2 Algorithm convergence analysis

The results of using the MFEAF algorithm in this article to calculate the anomaly prediction loss and fault classification prediction loss on training data are shown in Fig 2. As shown in the figure, with the increase of iteration times, the Anomaly Loss of the MFEAF algorithm gradually converges and approaches 0; The class loss of algorithm MFEAF quickly converges and tends towards 0. The experiment shows that the algorithm proposed in this paper can quickly converge in both anomaly prediction and fault classification prediction, demonstrating good convergence and effectively optimizing and capturing data patterns.

The time series prediction results using the MFEAF algorithm in this article are shown in Fig 3. This figure shows the comparison between the predicted results of time series data and the actual situation. The blue dashed line in the figure represents the True Line, which reflects the variation of actual time series data with timestamps. The red curve represents the prediction curve (Pred line), which reflects the algorithm's prediction of time series values at different timestamps. When the red predicted curve coincides with the blue true curve, it indicates that the model's prediction at the corresponding time point is highly consistent with the actual situation. As shown in the figure, it demonstrates the accuracy of the MFEAF algorithm in time series prediction, effectively capturing the changing patterns of time series data and providing reliable prediction results.

The results of anomaly detection using the MFEAF algorithm in this article are shown in Fig 4. The upper part of the figure demonstrates the anomaly detection capability of the model, visually presenting the model's prediction of the degree of data anomalies at different timestamps. The lower part presents the true abnormal situation, clearly reflecting the actual occurrence of the abnormality. By comparing the upper and lower parts, it is possible to effectively analyze the matching degree between the anomaly detection results of our algorithm and the real anomaly situation, indicating that our algorithm achieves good results in anomaly detection.

In the process of designing the network model, this study adopted a step-by-step design and validation strategy. The results showed that the complete MFEAF outperformed all variants in terms of performance indicators, indicating that each module contributed.

## 5.3 Comparative analysis of algorithms

To verify the performance of the algorithm proposed in this paper, performance analysis was conducted on six algorithms including DeepFT [9], TopoMAD [6], AWGG [5], PCFT [3], ECLB [7], and DFTM [2]. All baseline methods use

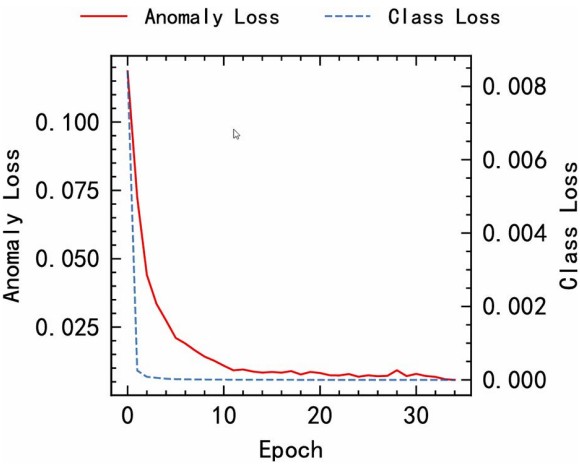

**Fig 2. Algorithm loss.**

**Fig 3. Algorithm Prediction Results.**

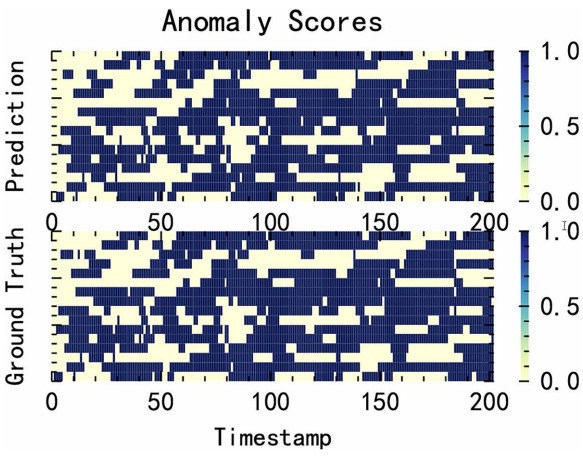

**Fig 4. Algorithm anomaly detection results.**

the hyperparameters recommended in their original paper and run in the same hardware and software environment. All methods are run 100 times, and the average and standard deviation are taken. To ensure the fairness of experimental conditions and consistency of experimental results, each baseline method adopts a training framework like the results of this paper, with the main difference being the model architecture and anomaly detection strategy of each baseline method itself. All methods share the same data preprocessing process.

(1)　Analysis of Fault Prediction Results

The comparative analysis of fault prediction results is shown in Table 1. MFEAF has a significant advantage over other algorithms, ranking first in accuracy (0.9436±0.0034), recall (0.9222±0.0162), and F1 score (0.9328±0.0066). Especially in terms of accuracy, it is about 9% higher than the suboptimal algorithms DeepFT (0.8635) and TopoMAD (0.8562).

(2)　Energy consumption analysis

The comparative analysis of energy consumption results is shown in Table 2. The MFEAF model has the lowest average energy consumption, at 14.69 kWh, which is superior to all other models. In contrast, the average energy consumption of

**Table 1. Comparative analysis of fault prediction results.**

| Algorithm | Accuracy (P) | Recall (R) | Comprehensive score (F1) |
|---|---|---|---|
| DeepFT | **0.8635±0.0011** | 0.9001±0.0092 | **0.8814±0.0271** |
| TopoMAD | **0.8562±0.0038** | 0.8927±0.0015 | **0.8741±0.0101** |
| AWGG | 0.8237±0.0124 | **0.9012±0.0081** | 0.8607±0.0135 |
| PCFT | 0.8029±0.0692 | **0.9018±0.0165** | 0.8495±0.0312 |
| ECLB | 0.7812±0.0711 | 0.8918±0.0203 | 0.8329±0.0901 |
| DFTM | 0.7713±0.0823 | 0.8427±0.0199 | 0.8054±0.0872 |
| **MFEAF** | **0.9436±0.0034** | **0.9222±0.0162** | **0.9328±0.0066** |

**Table 2. Comparative analysis of energy consumption results.**

| Algorithm | Total energy consumption | Average energy consumption | Interval energy consumption |
|---|---|---|---|
| DeepFT | 1809.70 | **15.47** | 17.92±0.23 |
| TopoMAD | 1807.75 | **15.32** | 17.90±0.25 |
| AWGG | 1853.54 | 15.98 | 18.35±0.24 |
| PCFT | 1882.38 | 16.23 | 18.64±0.26 |
| ECLB | 1771.82 | 16.56 | 17.54±0.19 |
| DFTM | 1834.33 | 16.68 | 18.16±0.21 |
| **MFEAF** | 1924.30 | **14.69** | 19.05±0.20 |

DeepFT is 15.47 kWh, which is about 5.30% higher than MFEAF. The average energy consumption of TopoMAD is 15.32 kWh, an increase of 4.30% compared to MFEAF. The average energy consumption of AWGG is 15.98 kWh, an increase of 8.11% compared to MFEAF, indicating a higher energy consumption. The average energy consumption of PCFT and ECLB is 16.23 kWh and 16.56 kWh, respectively. Their energy consumption increases by 10.89% and 12.74% compared to MFEAF, making them the two models with the highest energy consumption among all models. The average energy consumption of MFEAF is the lowest because it completes the highest number of tasks (131) among all algorithms (see Table 3), indicating that it achieves the lowest energy consumption while efficiently processing tasks.

(3)   Task processing capability analysis

The comparative analysis of the number of completed tasks is shown in Table 3. MFEAF not only has the highest total number of tasks completed, but also has the highest number of tasks completed in each interval, indicating that MFEAF is excellent in task allocation and load balancing. In contrast, DeepFT and TopoMAD showed a 9.92% and 10.69% decrease in task completion compared to MFEAF, respectively, and they exhibited significant fluctuations in interval task completion, resulting in relatively low efficiency. ECLB has the lowest number of tasks, reducing 24 tasks compared to MFEAF, with a decrease of 18.32%, indicating that its task processing ability is significantly insufficient in high load scenarios. The number of DFTM tasks decreased by 16.03%, which is better than ECLB but still significantly lagging behind MFEAF. AWGG and PCFT reduce 15 tasks compared to MFEAF, resulting in a decrease of 11.45%. Therefore, the MFEAF model performs the best in balancing task completion efficiency and load allocation, indicating that MFEAF has stronger task throughput capability in high load scenarios and is suitable for efficient task processing.

(4)   Average migration time

The comparative analysis of migration time is shown in Table 4. MFEAF achieved the best performance in average migration time (0.14±0.03), with the lowest time consumption among all comparative methods, demonstrating its efficient

**Table 3. Comparative Analysis of Task Completion Quantity.**

| Algorithm | Total number of completed tasks | The number of completed tasks within each interval |
|---|---|---|
| DeepFT | **117** | **1.16±0.16** |
| TopoMAD | **118** | **1.17±0.18** |
| AWGG | 116 | 1.15±0.18 |
| PCFT | 116 | 1.15±0.19 |
| ECLB | 107 | 1.06±0.17 |
| DFTM | 110 | 1.09±0.18 |
| **MFEAF** | **131** | **1.30±0.19** |

**Table 4. Comparative analysis of migration time.**

| Algorithm | Total migration time | Average migration time |
|---|---|---|
| DeepFT | 40.30 | 0.19±0.04 |
| AWGG | 29.95 | 0.20±0.07 |
| ECLB | 165.30 | 0.73±0.16 |
| DFTM | 35.77 | 0.29±0.07 |
| **MFEAF** | **19.79** | **0.14±0.03** |

and stable migration efficiency in edge dynamic environments. This advantage is attributed to MFEAF network model by integrating a graph attention network and a temporal network, which achieves accurate and efficient fault prediction and yields a better fault-tolerant scheduling optimization strategy, thereby significantly reducing the comprehensive cost of a single migration. This paper does not directly compare MFEAF with PCFT and TopoMAD in terms of migration efficiency, mainly due to the following considerations. PCFT is a virtual machine fault-tolerant coordination method for cloud computing environments, and its migration granularity and coordination mechanism are prone to causing a surge in migration times and durations in scenarios with limited edge resources and network fluctuations. TopoMAD relies on post-event fault detection results, and its passive recovery mode is prone to causing cascading migrations in dynamic edge environments, leading to a significant increase in migration costs.

(5) Number of task migrations

The comparative analysis of task migration times is shown in Table 5. The task migration frequency of MFEAF is only 160.00±0.22 times, significantly lower than other models. This indicates that its task scheduling algorithm can accurately match tasks and resources, reducing unnecessary migration. The average number of containers per interval for MFEAF is 82.62±0.03, with the highest quantity. This indicates that it can more efficiently utilize container resources, allocate tasks reasonably, make full use of containers within each interval, and avoid task migration caused by unreasonable resource allocation. Compared with the PCFT model with a task migration frequency of 834.00±0.33 times, the average number of containers per interval in PCFT is only 63.94±0.03, which may be due to poor container resource allocation and frequent task migration leading to low resource utilization efficiency. The average CPU utilization of MFEAF (see Table 8) is at a relatively high level among various models, and combined with fewer task migration times, it indicates that this model can maintain system stability while ensuring efficient utilization of CPU resources, without the need for frequent task migration to adjust resource allocation.

(6) Container cost

The container cost is calculated based on resource usage (CPU, memory, disk) and runtime, referring to the Alibaba Cloud container instance pricing model. The unit task cost is the total cost divided by the number of completed tasks.

   

**Table 5. Comparative Analysis of Task Migration Times.**

| Algorithm | Number of task migrations | The average number of containers per time interval |
|---|---|---|
| DeepFT | 238±0.24 | 69.75±0.03 |
| TopoMAD | 537±0.32 | 77.12±0.03 |
| AWGG | 187±0.18 | 75.56±0.03 |
| PCFT | 834±0.33 | 63.94±0.03 |
| ECLB | 345±0.31 | 77.25±0.02 |
| DFTM | 168±0.23 | 70.44±0.03 |
| **MFEAF** | **160±0.22** | **82.62±0.03** |

**Table 6. Comparative analysis of container costs.**

| Algorithm | Container cost (yuan) | Unit task cost (yuan/task) |
|---|---|---|
| DeepFT | **6.69** | **3.99** |
| TopoMAD | **6.63** | 4.34 |
| AWGG | 6.74 | 4.39 |
| PCFT | 6.74 | **3.71** |
| ECLB | 7.31 | 5.28 |
| DFTM | 7.11 | 4.56 |
| **MFEAF** | **5.97** | **3.76** |

As shown in Table 6, the comparative analysis of container costs shows that MFEAF has the lowest single container cost (5.97 yuan), saving 11% to 22% of costs compared to other models. MFEAF has the best unit task cost (3.76 yuan/task) and the highest resource utilization rate (see Table 8), indicating that it has reduced unit task costs through efficient resource allocation. Its low migration frequency (160 times) (see Table 5) and short migration time (19.79 seconds) (see Table 4) further reduce additional expenses. ECLB has the highest cost (+22.45%) and the least number of tasks (see Table 3), while PCFT has slightly lower unit task costs, but significant disadvantages in migration time (see Table 4) and task count (see Table 3). TopoMAD and DeepFT have moderate costs (+11%~12%), and there is a large gap in task count compared to MFEAF (−10%).

(7) Fairness (Jain Index)

To evaluate the fairness of scheduling algorithms in resource allocation, this paper adopts Jain's Fairness Index as the quantitative standard. Fairness specifically refers to the degree of balance in the instantaneous computing throughput (IPS, Instructions Per Second) obtained by active containers at each time step. IPS is mapped from CPU usage rates in real workload trajectories and represents the actual computing power that containers can execute. The Jain Index measures whether the scheduler allocates limited computing resources reasonably and impartially among multiple concurrent containers. The calculation method of this index is given by formula (31). $x_i$ represents the IPS value obtained by the i-th container at a specific time step, and n is the number of active containers. The value range of the Jain Index is $\left[\frac{1}{n}, 1\right]$, with a value closer to 1 indicating a more equitable allocation.

$$F(x_1, x_2, ..., x_n) = \frac{(\sum_{i=1}^{n} x_i)^2}{n \cdot \sum_{i=1}^{n} x_i^2}$$

(31)

In the edge computing task scheduling scenario, the role of this metric is to transform abstract fairness into comparable numerical values through standardized calculation, objectively measuring the distribution balance of computing resources obtained by each host or container. The importance of this metric lies in its close correlation with system stability, energy efficiency, and quality of service (QoS): fair resource allocation can effectively prevent node overload, reduce the overhead of fault migration, and avoid task starvation, thereby enhancing overall system performance. In terms of relevance, this metric is particularly suitable for edge computing dynamic scheduling scenarios, as it can not only reflect the load balancing status in real time but also form a complete scheduling strategy evaluation system together with metrics such as task completion quantity, energy consumption, and migration time, providing a key basis for algorithm optimization and selection.

The comparative analysis of fairness (Jain index) is shown in Table 7. MFEAF has the highest fairness (0.55), leading other models by 5.45%~20%, indicating that its resource allocation strategy is the most balanced. PCFT performed the worst (0.44), 20% lower than MFEAF, possibly due to frequent migration or unfair scheduling strategies leading to intense resource competition. TopoMAD has a similar fairness to AWGG (0.52), which is 5.45% lower than MFEAF. The standard deviation of the Jain index for all models is ±0.02, indicating a consistent range of fairness fluctuations. In addition, the high fairness (0.55) of MFEAF combined with high task volume (131) (see Table 3) and low migration time (19.79 seconds) (see Table 4) indicates that it can efficiently allocate resources and avoid performance degradation caused by resource competition.

(8)  Average CPU utilization

The comparative analysis of average CPU utilization is shown in Table 8. MFEAF has the highest average CPU utilization rate (61.09%), leading other models by 3.63%~13.60%, indicating that its resource scheduling strategy can maximize the utilization of computing resources. ECLB performed the worst (47.49%), with a utilization rate 13.60% lower than MFEAF, and prominent resource idle issues, possibly due to uneven task allocation or excessive migration costs. TopoMAD and DeepFT have similar utilization rates (50.79% and 50.73%), which are about 10.3% lower than MFEAF. Overall, the high CPU utilization of MFEAF (61.09%) combined with its high task volume (131 tasks) (see Table 3) and low energy consumption (14.69 kWh) (see Table 2) indicates that it achieves a balance between performance and energy efficiency through efficient resource allocation.

(9)  Result analysis

All algorithms were independently run 100 times in the same experimental environment to collect statistical data on key performance indicators. In algorithm comparative analysis, the value following "±" represents the half width of the 90% confidence interval. This reflects the fluctuation range of the indicator over multiple runs and is used to evaluate the stability of the results. Taking the confidence interval analysis of fault prediction, energy consumption, and task completion

**Table 7. Comparative Analysis of Equity (Jain Index).**

| Algorithm | DeepFT | TopoMAD | AWGG | PCFT | ECLB | DFTM | MFEAF |
|---|---|---|---|---|---|---|---|
| Fairness (Jain Index) | 0.47±0.02 | 0.52±0.02 | 0.52±0.02 | 0.44±0.02 | 0.50±0.02 | 0.49±0.02 | **0.55±0.02** |

**Table 8. Comparative Analysis of Average CPU Utilization.**

| Algorithm | DeepFT | TopoMAD | AWGG | PCFT | ECLB | DFTM | MFEAF |
|---|---|---|---|---|---|---|---|
| Average CPU utilization rate (%) | 50.73 ±2.01 | 50.79 ±2.20 | 54.62 ±2.16 | 57.46 ±2.31 | 47.49 ±1.68 | 52.99 ±1.88 | **61.09 ±1.77** |

quantity as an example, the statistical information will be analyzed. In the comparative analysis of the fault prediction results in Table 2, the lower confidence interval (0.9262) of the F1 score (0.9328±0.0066) of MFEAF is higher than the upper limit (0.9085) of the suboptimal algorithm DeepFT, indicating that the performance improvement is statistically significant. In the comparative analysis of energy consumption results in Table 2, the confidence interval for the average energy consumption of MFEAF (14.69 kWh) does not overlap with DeepFT (15.47 kWh) and TopoMAD (15.32 kWh), indicating a significant energy efficiency advantage. In the comparative analysis of task completion numbers in Table 6, the confidence intervals of MFEAF's total task count (131) and its interval task count (1.30±0.19) are higher than other algorithms, indicating that its task processing ability is significantly better (with no overlapping intervals).

The factors that affect the scalability of the algorithm in this article mainly include the number of tasks, the number of hosts, and the timing length. In terms of task scalability, the algorithm proposed in this paper exhibits a linear increase in task scalability. In terms of host scalability, complexity can be reduced through group attention or local perceptual fields. In terms of temporal length scalability, the results of this paper show a linear increase in task number scalability.

## 6 Conclusion

To address the challenges of fault prediction and fault-tolerant scheduling caused by resource contention in resource-constrained edge computing networks, this paper proposes an edge computing task scheduling mechanism based on multi-dimensional feature extraction and attention fusion. MFEAF integrates Graph Attention Networks, Graph Convolutional Networks, Long Short-Term Memory networks, and Residual Networks to achieve accurate system-state prediction and dynamic dependency modeling. A model training algorithm is designed to optimize network parameters, and a scheduling optimization algorithm is developed to adjust scheduling decisions. Experimental results demonstrate that MFEAF significantly alleviates the problems of low task completion rates and high host energy consumption caused by limited device resources in edge computing environments, thereby enhancing system stability and resource utilization.

Although MFEAF demonstrates good performance in edge computing task scheduling, there are still several directions worth further exploration. Firstly, at the algorithm level, research can be conducted on lighter graph neural network structures to reduce computational overhead and adapt to edge devices with more limited resources. Secondly, in terms of application scenarios, MFEAF can be extended to areas with high requirements for real-time performance and reliability, such as intelligent transportation systems, industrial IoT, and smart healthcare, and customized optimization can be carried out in combination with domain knowledge. Finally, the integration with technologies such as federated learning and blockchain can be explored to achieve secure and trusted task scheduling and resource management in a distributed edge environment.

## Supporting information

**S1 File. Data.**
(ZIP)

## Author contributions

**Conceptualization:** shunli zhang, Peng Yu.

**Data curation:** shunli zhang.

**Formal analysis:** shunli zhang.

**Funding acquisition:** shunli zhang.

**Investigation:** shunli zhang.

**Methodology:** shunli zhang, Peng Yu.

**Project administration:** shunli zhang.

**Resources:** shunli zhang, Peng Yu.

**Software:** shunli zhang, Jia-ying Li.

**Supervision:** shunli zhang.

**Validation:** shunli zhang, Jia-ying Li.

**Visualization:** shunli zhang, Jia-ying Li.

**Writing – original draft:** shunli zhang, Jia-ying Li.

**Writing – review & editing:** shunli zhang, Jia-ying Li.

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
