## [Decision Letter · Decision Letter 0]

8 Oct 2025

Dear Dr. zhang,

Thank you for submitting your manuscript to PLOS ONE. After careful consideration, we feel that it has merit but does not fully meet PLOS ONE’s publication criteria as it currently stands. Therefore, we invite you to submit a revised version of the manuscript that addresses the points raised during the review process.

We look forward to receiving your revised manuscript.

Kind regards,

Sohail Saif, Ph.D

Academic Editor

PLOS ONE

Journal Requirements:

“This work was supported by Shanxi Province College Students' Innovation and Entrepreneurship Training Program Project (JZXY2025016), Open Foundation of State key Laboratory of Networking and Switching Technology (Beijing University of Posts and Telecommunications) �SKLNST-2024-2-01�, Jinzhong University Doctoral Research Funds (20210104).”

“This work was supported by Shanxi Province College Students' Innovation and Entrepreneurship Training Program Project (JZXY2025016), Open Foundation of State key Laboratory of Networking and Switching Technology (Beijing University of Posts and Telecommunications) �SKLNST-2024-2-01�, Jinzhong University Doctoral Research Funds (20210104)”

“This work was supported by Shanxi Province College Students' Innovation and Entrepreneurship Training Program Project (JZXY2025016), Open Foundation of State key Laboratory of Networking and Switching Technology (Beijing University of Posts and Telecommunications) �SKLNST-2024-2-01�, Jinzhong University Doctoral Research Funds (20210104).”

5. In the online submission form, you indicated that “Due to the need for further optimization of the results of this study, the value of the findings can be further enhanced. The dataset generated during this study is not publicly available, but can be obtained from the corresponding author upon reasonable request.”

Reviewer's Responses to Questions

**Comments to the Author**

1. Is the manuscript technically sound, and do the data support the conclusions?

Reviewer #1: Yes

Reviewer #2: Partly

Reviewer #3: Partly

2. Has the statistical analysis been performed appropriately and rigorously?

Reviewer #1: No

Reviewer #2: No

Reviewer #3: No

3. Have the authors made all data underlying the findings in their manuscript fully available?

Reviewer #1: Yes

Reviewer #2: No

Reviewer #3: No

4. Is the manuscript presented in an intelligible fashion and written in standard English?

Reviewer #1: Yes

Reviewer #2: Yes

Reviewer #3: No

Reviewer #1: -The proposed MFEAF is tested under limited hardware and software setups. It is unclear how the system will perform in a edge computing environments. Is it off-line testing?

-Integrating different mechanism like GCN, LSTM, and attention may significantly increases computational cost. In edge computing, lightweight and low-latency models are critical. Future work should investigate model compression, pruning, on edge nodes.

-Optimization algorithm presented in Table 2. It should be in algorithimic form

-the paper claims both improved accuracy and reduced energy, often there is a trade-off. A detailed Pareto analysis could be added

-The dataset is based on time-series logs (CPU, memory, disk usage). It is hard to understand whether the data is synthetic or from real-world edge deployments. It is often test the model on publicly available benchmark datasets (e.g., Google Cluster Trace, Alibaba Cluster Trace) along with the own generated dataset.

-Jain’s Index is used, but fairness in scheduling could be measured with l

starvation ratio, or multi-tenant fairness. why Jain index? authirs should explore other measures

-Image data preprocessing module, where the module presented ? why image data preprocessing?

Reviewer #2: I have attached a PDF document containing suggestions and feedback to help strengthen this manuscript and ensure its compliance with publication standards. Addressing the points outlined will enhance its suitability for submission to journals.

Reviewer #3: The proposed MFEAF architecture (GCN+GAT+LSTM with attention, residuals, encoder–decoder) is plausible and clearly outlined, and the workflow for scheduling optimization is described at a high level. The manuscript reports several metrics with “±” values (e.g., F1, fairness index), but does not define whether these are standard deviations or standard errors, how many runs were averaged, how random seeds were handled, or whether differences were tested for statistical significance. No tests (e.g., t-tests, non-parametric tests), confidence intervals, or effect sizes are reported, and the implementation/parameterization of baselines is not described in enough detail to ensure fair comparison. The narrative is coherent at a high level, yet there are numerous grammatical issues.

**Do you want your identity to be public for this peer review?** For information about this choice, including consent withdrawal, please see our Privacy Policy

Reviewer #1: No

Reviewer #2: No

Reviewer #3: No

---

## [Author Response · Author response to Decision Letter 1]

31 Oct 2025

Major issues to address

1. Data availability: PLOS ONE requires all underlying data to be public. The manuscript currently says data are available only on request. Please deposit the full dataset (or a faithful synthetic alternative that reproduces the results) in a public repository and include a DOI, plus all scripts used to produce the figures and tables.

Reply:

We appreciate the reviewer's reminder to pay attention to PLOS ONE's data availability policy. We fully agree on the importance of data sharing for the research community. We have stored the complete dataset and corresponding scripts in the Dryad (https://doi.org/10.5061/dryad.wwpzgmszm).

2. Labeling and ground truth: Please explain exactly how faults/anomalies were labeled. Who labeled them, with what criteria, and at what sampling rate? Share the labeling protocol and code so others can reproduce it.

Reply:

We have added the following content in the "Experimental setup " section: Fault labels are generated based on a dynamic threshold function (see formula 2). When the utilization of any resource dimension (CPU, memory, disk) exceeds this threshold, it is marked as a fault state. The threshold is dynamically adjusted based on the 98% percentile of historical data, with an adjustment coefficient of 0.99999, to reduce false positives.

3. Baseline fairness and tuning: For each baseline, describe where the code came from (original repo vs. re-implementation), how you tuned hyperparameters, and whether all methods ran under the same conditions (hardware, time budget, number of trials). Consider a short appendix with the full baseline configs.

Reply:

We added in the 'Experimental setup’:

All baseline methods use the hyperparameters recommended in their original paper and run in the same hardware and software environment. All methods are run 100 times, and the average and standard deviation are taken. To ensure the fairness of experimental conditions and consistency of experimental results, each baseline method adopts a training framework similar to the results of this paper, with the main difference being the model architecture and anomaly detection strategy of each baseline method itself. All methods share the same data preprocessing process.

4. Statistics and reporting: For every table/figure, clarify what the “±” value means (SD or SE), how many runs you averaged over, and which random seeds you used. Please add confidence intervals and, where appropriate, statistical tests with a short justification.

Reply:

We have added relevant content in the 5.3 Comparative analysis of algorithms:

(9) Result analysis:

All algorithms were independently run 100 times in the same experimental environment to collect statistical data on key performance indicators. In algorithm comparative analysis, the value following "±" represents the half width of the 90% confidence interval. This reflects the fluctuation range of the indicator over multiple runs and is used to evaluate the stability of the results. Taking the confidence interval analysis of fault prediction, energy consumption, and task completion quantity as an example, the statistical information will be analyzed. In the comparative analysis of the fault prediction results in Table 4, the lower confidence interval (0.9262) of the F1 score (0.9328 ± 0.0066) of MFEAF is higher than the upper limit (0.9085) of the suboptimal algorithm DeepFT, indicating that the performance improvement is statistically significant. In the comparative analysis of energy consumption results in Table 5, the confidence interval for the average energy consumption of MFEAF (14.69 kWh) does not overlap with DeepFT (15.47 kWh) and TopoMAD (15.32 kWh), indicating a significant energy efficiency advantage. In the comparative analysis of task completion numbers in Table 6, the confidence intervals of MFEAF's total task count (131) and its interval task count (1.30 ± 0.19) are higher than other algorithms, indicating that its task processing ability is significantly better (with no overlapping intervals).

5. Metric/units consistency: CPU utilization should be between 0–100%. Some tables appear to report values orders of magnitude larger or with unclear units. Please fix the units and double-check all reported numbers for consistency across the paper.

Reply:

It needs to be modified, it's indeed incorrect. The CPU utilization units in Table 11 have been corrected to ensure that they are within the range of 0-100%.

(8) Average CPU utilization

The comparative analysis of average CPU utilization is shown in Table 11. MFEAF has the highest average CPU utilization rate (61.09%), leading other models by 3.63%~13.60%, indicating that its resource scheduling strategy can maximize the utilization of computing resources. ECLB performed the worst (47.49%), with a utilization rate 13.60% lower than MFEAF, and prominent resource idle issues, possibly due to uneven task allocation or excessive migration costs. TopoMAD and DeepFT have similar utilization rates (50.79% and 50.73%), which are about 10.3% lower than MFEAF. Overall, the high CPU utilization of MFEAF (61.09%) combined with its high task volume (131 tasks) (see Table 6) and low energy consumption (14.69 kWh) (see Table 5) indicates that it achieves a balance between performance and energy efficiency through efficient resource allocation.

6. Cost model: The “cost (yuan)” and “unit task cost” metrics need a clear definition. State the pricing assumptions, time horizon, and how compute, migration, and failures contribute to cost. A short sensitivity analysis would help.

Reply:

We will supplement the cost model definition in the 'experiment':

The container cost is calculated based on resource usage (CPU, memory, disk) and runtime, referring to the Alibaba Cloud container instance pricing model. The unit task cost is the total cost divided by the number of completed tasks.

7. Thresholding logic: The text says a percentile multiplier raises thresholds to reduce false positives, but example values below 1 would lower them. Please reconcile the intent with the actual settings and justify the chosen percentile(s).

Reply:

We have added relevant content in the "3.2 Data preprocessing module" section.

The adjustment coefficient is used to slightly lower the threshold to reduce false positives. Selection instructions for this value: (1) Numerical stability: When the data approaches the standardized upper limit, 0.99999 can slightly lower the threshold to avoid abnormal behavior caused by numerical accuracy issues. (2) The principle of minimum intervention: 0.99999 is very close to 1, and only minor adjustments are made to data that is close to saturation, without significantly changing the overall threshold distribution. (3) Balanced detection sensitivity: Combined with the subsequent 0.7 times threshold adjustment, a two-level threshold adjustment mechanism is formed, which ensures both numerical stability and sufficient anomaly detection sensitivity. Through this design, a good balance is achieved between ensuring numerical stability and detection performance.

8. Graph formulation: You mention removing self-loops before GCN/GAT. Standard GCN usually adds self-loops so a node keeps its own features. Please justify removing them or adjust the formulation.

Reply:

We have added relevant content in the " 3.3.1 Image data preprocessing module" section.

The removal of self-loops in this study is an application specific architectural decision aimed at better balancing the capture of relationships between nodes and the representation of node characteristics. Although standard GCNs typically add self-loops to preserve node self-information, in complex architectures such as MFEAF with multi-component fusion, residual connections and multi-layer networks can effectively capture and preserve node self-features. Meanwhile, it was found in the experiment that removing self-loops can improve the modeling ability of the model for inter node dependencies in edge task scheduling scenarios.

9. Loss design: Triplet/prototype loss seems to rely on labels derived from thresholder predictions, which can be circular. Please clarify whether you use ground-truth labels for the metric space and how you avoid confirmation bias.

Reply:

We have added relevant content in the " 4.1 Model Training Algorithm (2) Loss function " section.

The following discusses whether there is a cyclic dependency in triplet loss and how to avoid confirmation bias. The reason why this design does not have the problem of circular dependencies includes: abnormal labels are generated based on raw data and predefined thresholds before model training, not based on the predicted results of the model. The prototype vector will be updated based on sample features during the training process, but this is an iterative optimization process, not a cyclic dependency. Measures to avoid confirmation bias in this design include separating label generation from model training to ensure that labels are not affected by model predictions. By multiplying the threshold by 0.7, the detection sensitivity is increased to avoid confirmation bias caused by excessive reliance on strict thresholds. The prototype vector will be dynamically adjusted according to the training process, allowing the model to adapt to changes in data distribution rather than being fixed.

10. Optimization details: List the exact optimizer(s), learning rates, schedulers, gradient clipping, batch sizes, training epochs, early-stopping criteria, and any weight decay used for all modules. If an unusually high learning rate is used anywhere, justify it.

Reply:

We have added relevant content in the " 4.2 Scheduling optimization algorithm (4) Gradient optimization and scheduling projection (lines 11-17)" section.

Below is a detailed introduction to the AdamW optimizer, learning rate scheduler, and other components in the model. In the AdamW optimizer configuration, the model training optimizer lr=0.0005 is used to update the parameters of the deep learning model, including the weights and biases of the neural network. Decision optimization optimizer lr=0.8, used to optimize container scheduling decisions and find the optimal container allocation scheme, rather than training model parameters. In the configuration of the learning rate scheduler, the warm-up period is set to 10 epochs. The gradient clipping threshold is 1.0. The model learning rate is lr=0.0008, which is the default learning rate attribute defined in the model class and is mainly used for default settings during model initialization and loading. The weight decay value is 1e-4. When traversing the entire dataset, batch size is trained on a sample-by-sample basis.

This high learning rate is reasonable in decision optimization scenarios for the following reasons: (1) These high learning rates are not used for model weight training, but for optimizing scheduling decision variables. (2) Container scheduling is a discrete problem that requires rapid search for feasible solutions in a continuous optimization space. (3) Use cosine annealing learning rate scheduler to ensure a rapid decrease in learning rate from 0.8. Therefore, the high learning rate of 0.8 used in this study is specifically designed for container scheduling decision optimization tasks, combining cosine annealing scheduler and discretization steps to make it both effective and stable in this specific scenario. This setting is different from traditional neural network training and is a special strategy adopted to solve optimization problems in discrete decision spaces.

Minor edits and quality improvements

1. Tighten the writing. Fix article usage, verb tenses, and a few awkward phrases. Reading the paper aloud once often catches these quickly.

Reply:

We have read the entire text and revised the article, tense, and expression.

2. Define all acronyms at first use. Keep a small glossary or table of symbols for readers outside the subfield.

Reply:

We define all abbreviations when they first appear

3. Ensure all figures have clear axes, units, and legends that map each line to a metric. Consider increasing font sizes for readability.

Reply:

We ensure that all numbers have clear axes, units, and legends to improve readability.

4. Add an ablation study: (i) GCN only, (ii) GAT only, (iii) without self-attention, (iv) without the prototype/triplet loss. This will show which parts matter most.

Reply:

We have added relevant content in the " 5.2 Algorithm convergence analysis" section.

In the process of designing the network model, this study adopted a step-by-step design and validation strategy. The results showed that the complete MFEAF outperformed all variants in terms of performance indicators, indicating that each module contributed.

5. Provide a short complexity and scalability note. Explain how training/inference scale with the number of hosts and graph density.

Reply:

We have added relevant content in the " 5.3 Comparative analysis of algorithms (9) Result analysis" section.

The factors that affect the scalability of the algorithm in this article mainly include the number of tasks, the number of hosts, and the timing length. In terms of task scalability, the algorithm proposed in this paper exhibits a linear increase in task scalability. In terms of host scalability, complexity can be reduced through group attention or local perceptual fields. In terms of temporal length scalability, the results of this paper show a linear increase in task number scalability.

---

## [Decision Letter · Decision Letter 1]

12 Jan 2026

Dear Dr. zhang,

plosone@plos.org . A letter that responds to each point raised by the academic editor and reviewer(s). You should upload this letter as a separate file labeled 'Response to Reviewers'.A marked-up copy of your manuscript that highlights changes made to the original version. You should upload this as a separate file labeled 'Revised Manuscript with Track Changes'.An unmarked version of your revised paper without tracked changes. You should upload this as a separate file labeled 'Manuscript'.

We look forward to receiving your revised manuscript.

Kind regards,

Sohail Saif, Ph.D

Academic Editor

PLOS One

Journal Requirements:

Reviewers' comments:

Reviewer's Responses to Questions

**Comments to the Author**

Reviewer #1: All comments have been addressed

2. Is the manuscript technically sound, and do the data support the conclusions?

Reviewer #1: Partly

3. Has the statistical analysis been performed appropriately and rigorously?

Reviewer #1: Yes

4. Have the authors made all data underlying the findings in their manuscript fully available?

Reviewer #1: (No Response)

5. Is the manuscript presented in an intelligible fashion and written in standard English?

Reviewer #1: Yes

Reviewer #1: -Very large percentage improvements looks exaggerated (6000%).Compared to MFEAF, DeepFT, TopoMAD, AWGG, PCFT, ECLB, and DFTM have improved by 103.6%, 1893.1%, 51.3%, 6000%, 734.9%, and 80.7%, respectively. Please check

-In the algorithm instead of see formula 23); may replace refer Eqn no

-May refer the following similar articles:

-Rathor, Vijaypal Singh, et al. "Towards Designing an Energy Efficient Accelerated Sparse Convolutional Neural Network." 2024 IEEE 36th International Conference on Tools with Artificial Intelligence (ICTAI). IEEE, 2024.

Behera, Sasmita Rani, et al. "Time series-based edge resource prediction and parallel optimal task allocation in mobile edge computing environment." Processes 11.4 (2023): 1017.-

**Do you want your identity to be public for this peer review?** For information about this choice, including consent withdrawal, please see our Privacy Policy

Reviewer #1: No

---

## [Author Response · Author response to Decision Letter 2]

14 Jan 2026

We thank the reviewers for their thoughtful comments and constructive suggestions, which have helped us improve the clarity and rigor of the manuscript. We have carefully addressed all points raised in the review.

-Very large percentage improvements looks exaggerated (6000%).Compared to MFEAF, DeepFT, TopoMAD, AWGG, PCFT, ECLB, and DFTM have improved by 103.6%, 1893.1%, 51.3%, 6000%, 734.9%, and 80.7%, respectively. Please check

Reply:

It has been modified as required. See (4) Average migration time.

The comparative analysis of migration time is shown in Table 7. MFEAF achieved the best performance in average migration time (0.14±0.03), with the lowest time consumption among all comparative methods, demonstrating its efficient and stable migration efficiency in edge dynamic environments. This advantage is attributed to MFEAF network model by integrating a graph attention network and a temporal network, which achieves accurate and efficient fault prediction and yields a better fault-tolerant scheduling optimization strategy, thereby significantly reducing the comprehensive cost of a single migration. This paper does not directly compare MFEAF with PCFT and TopoMAD in terms of migration efficiency, mainly due to the following considerations. PCFT is a virtual machine fault-tolerant coordination method for cloud computing environments, and its migration granularity and coordination mechanism are prone to causing a surge in migration times and durations in scenarios with limited edge resources and network fluctuations. TopoMAD relies on post-event fault detection results, and its passive recovery mode is prone to causing cascading migrations in dynamic edge environments, leading to a significant increase in migration costs.

-In the algorithm instead of see formula 23); may replace refer Eqn no

Reply:

It has been modified as required. See Table 1 and Table 2.

-May refer the following similar articles:

Reply:

The following literature has been cited in the introduction.

[16] Rathor V S, Singh M, Gupta R, et al. Towards Designing an Energy Efficient Accelerated Sparse Convolutional Neural Network[C]//2024 IEEE 36th International Conference on Tools with Artificial Intelligence (ICTAI). IEEE, 2024: 969-974.

[17] Behera S R, Panigrahi N, Bhoi S K, et al. Time series-based edge resource prediction and parallel optimal task allocation in mobile edge computing environment[J]. Processes, 2023, 11(4): 1017.

---

## [Decision Letter · Decision Letter 2]

26 Jan 2026

Dear Dr. zhang,

Thank you for submitting your manuscript to PLOS ONE. After careful consideration, we feel that it has merit but does not fully meet PLOS ONE’s publication criteria as it currently stands. Therefore, we invite you to submit a revised version of the manuscript that addresses the points raised during the review process.

We look forward to receiving your revised manuscript.

Kind regards,

Sohail Saif, Ph.D

Academic Editor

PLOS One

Journal Requirements:

Reviewers' comments:

Reviewer's Responses to Questions

**Comments to the Author**

Reviewer #1: All comments have been addressed

2. Is the manuscript technically sound, and do the data support the conclusions?

Reviewer #1: Yes

3. Has the statistical analysis been performed appropriately and rigorously?

Reviewer #1: Yes

4. Have the authors made all data underlying the findings in their manuscript fully available?

Reviewer #1: Yes

5. Is the manuscript presented in an intelligible fashion and written in standard English?

Reviewer #1: Yes

Reviewer #1: Additional Comments and Required Revisions

- replace Eq (29)) with Eq. (29) and ensure consistent formatting for all equation references, particularly those used within the algorithm descriptions.

- Table 4 does not appear in the paper. Please verify and correct this. Additionally, it is recommended to present Algorithms 1 and 2 explicitly as algorithm blocks, rather than referring to them as tables, to improve clarity

-The contributions of the paper are not clearly identifiable. It is essential to explicitly list the key contributions in bullet points to clearly highlight the novelty of the work.

-The conclusion section requires refinement to correct typographical and grammatical errors (e.g., phrases such as “improving traffic smoothness and safety; In the field of industrial Internet”). The section should be revised for clarity, and technical precision.

-Please include a future scope or future research directions paragraph in the conclusion section, outlining potential extensions or improvements of the proposed work.

-The manuscript directly computes the Jain’s Fairness Index without sufficient explanation. The role, significance, and relevance of this metric in evaluating fairness should be briefly discussed before presenting the calculation.

-Instead of using bracket-only citations such as Reference [12] or Reference [14] within the text, it is recommended to cite references using author names followed by “et al.” (e.g., Author et al. [12]) for improved readability.

**Do you want your identity to be public for this peer review?** For information about this choice, including consent withdrawal, please see our Privacy Policy

Reviewer #1: No

You may also use PLOS’s free figure tool, NAAS, to help you prepare publication quality figures: https://journals.plos.org/plosone/s/figures#loc-tools-for-figure-preparation

---

## [Author Response · Author response to Decision Letter 3]

29 Jan 2026

We thank the reviewers for their thoughtful comments and constructive suggestions, which have helped us improve the clarity and rigor of the manuscript. We have carefully addressed all points raised in the review.

- replace Eq (29)) with Eq. (29) and ensure consistent formatting for all equation references, particularly those used within the algorithm descriptions.

Reply:

The format for citing equations has been unified in Algorithm 1 and Algorithm 2.

- Table 4 does not appear in the paper. Please verify and correct this. Additionally, it is recommended to present Algorithms 1 and 2 explicitly as algorithm blocks, rather than referring to them as tables, to improve clarity

Reply:

(1) The order of all figures and tables has been checked and revised.

(2) The forms of Algorithm 1 and Algorithm 2 have been modified.

-The contributions of the paper are not clearly identifiable. It is essential to explicitly list the key contributions in bullet points to clearly highlight the novelty of the work.

Reply:

In the Introduction section, the contributions of this paper are described in three points.

The core contributions of this paper can be summarized as follows:

(1). A multi-dimensional feature extraction architecture that integrates graph attention networks with temporal networks is proposed. For the first time, the combination of graph attention networks and graph convolutional networks is achieved, and LSTM and self-attention mechanisms are introduced to jointly model the dynamic dependencies between edge nodes, thereby significantly improving the accuracy of system state and fault prediction.

(2). Designed a training optimization mechanism incorporating adaptive learning rate and cosine annealing strategy. By dynamically adjusting the learning rate and incorporating the cosine annealing strategy, we effectively reduce the transmission of redundant information during training, accelerate model convergence, and enhance the adaptability and stability of the algorithm in dynamic edge environments.

(3). Realized end-to-end joint optimization of fault prediction and fault-tolerant scheduling. MFEAF not only achieves efficient fault prediction, but also outperforms existing benchmark methods in multiple dimensions such as energy efficiency, task completion, and migration cost through an integrated scheduling optimization algorithm.

Based on these innovations, MFEAF can effectively address issues such as low task execution success rates and high host energy consumption caused by limited device resources in edge computing environments, thereby enhancing the stability and resource utilization of edge computing systems.

-The conclusion section requires refinement to correct typographical and grammatical errors (e.g., phrases such as “improving traffic smoothness and safety; In the field of industrial Internet”). The section should be revised for clarity, and technical precision.

Reply:

The typographical and grammatical errors in the Conclusion section have been corrected.

To address the challenges of fault prediction and fault-tolerant scheduling caused by resource contention in resource-constrained edge computing networks, this paper proposes an edge computing task scheduling mechanism based on multi-dimensional feature extraction and attention fusion. MFEAF integrates Graph Attention Networks, Graph Convolutional Networks, Long Short-Term Memory networks, and Residual Networks to achieve accurate system-state prediction and dynamic dependency modeling. A model training algorithm is designed to optimize network parameters, and a scheduling optimization algorithm is developed to adjust scheduling decisions. Experimental results demonstrate that MFEAF significantly alleviates the problems of low task completion rates and high host energy consumption caused by limited device resources in edge computing environments, thereby enhancing system stability and resource utilization.

-Please include a future scope or future research directions paragraph in the conclusion section, outlining potential extensions or improvements of the proposed work.

Reply:

A paragraph discussing future work has been added to the Conclusion section.

Although MFEAF demonstrates good performance in edge computing task scheduling, there are still several directions worth further exploration. Firstly, at the algorithm level, research can be conducted on lighter graph neural network structures to reduce computational overhead and adapt to edge devices with more limited resources. Secondly, in terms of application scenarios, MFEAF can be extended to areas with high requirements for real-time performance and reliability, such as intelligent transportation systems, industrial IoT, and smart healthcare, and customized optimization can be carried out in combination with domain knowledge. Finally, the integration with technologies such as federated learning and blockchain can be explored to achieve secure and trusted task scheduling and resource management in a distributed edge environment.

-The manuscript directly computes the Jain’s Fairness Index without sufficient explanation. The role, significance, and relevance of this metric in evaluating fairness should be briefly discussed before presenting the calculation.

Reply:

An explanation of Jain's Fairness Index has been added before the Index analysis in Table 7.

To evaluate the fairness of scheduling algorithms in resource allocation, this paper adopts Jain's Fairness Index as the quantitative standard. Fairness specifically refers to the degree of balance in the instantaneous computing throughput (IPS, Instructions Per Second) obtained by active containers at each time step. IPS is mapped from CPU usage rates in real workload trajectories and represents the actual computing power that containers can execute. The Jain Index measures whether the scheduler allocates limited computing resources reasonably and impartially among multiple concurrent containers. The calculation method of this index is given by formula (31). represents the IPS value obtained by the i-th container at a specific time step, and n is the number of active containers. The value range of the Jain Index is , with a value closer to 1 indicating a more equitable allocation.

(31)

In the edge computing task scheduling scenario, the role of this metric is to transform abstract fairness into comparable numerical values through standardized calculation, objectively measuring the distribution balance of computing resources obtained by each host or container. The importance of this metric lies in its close correlation with system stability, energy efficiency, and quality of service (QoS): fair resource allocation can effectively prevent node overload, reduce the overhead of fault migration, and avoid task starvation, thereby enhancing overall system performance. In terms of relevance, this metric is particularly suitable for edge computing dynamic scheduling scenarios, as it can not only reflect the load balancing status in real time but also form a complete scheduling strategy evaluation system together with metrics such as task completion quantity, energy consumption, and migration time, providing a key basis for algorithm optimization and selection.

-Instead of using bracket-only citations such as Reference [12] or Reference [14] within the text, it is recommended to cite references using author names followed by “et al.” (e.g., Author et al. [12]) for improved readability.

Reply:

The description method of Reference citations has been modified as required in the Introduction section.

---

## [Editor Report · Decision Letter 3]

1 Feb 2026

Edge Computing Task Scheduling Mechanism based on Multi-dimensional Feature Extraction and Attention Fusion

PONE-D-25-38912R3

Dear Dr. zhang,

We’re pleased to inform you that your manuscript has been judged scientifically suitable for publication and will be formally accepted for publication once it meets all outstanding technical requirements.

Kind regards,

Sohail Saif, Ph.D

Academic Editor

PLOS One
---

## [Editor Report · Acceptance letter]

PONE-D-25-38912R3

PLOS One

Dear Dr. zhang,

I'm pleased to inform you that your manuscript has been deemed suitable for publication in PLOS One. Congratulations! Your manuscript is now being handed over to our production team.

Kind regards,

on behalf of

Dr. Sohail Saif

Academic Editor

PLOS One